# Targeting Opioid Receptors in Addiction and Drug Withdrawal: Where Are We Going?

**DOI:** 10.3390/ijms241310888

**Published:** 2023-06-29

**Authors:** Rita Tabanelli, Simone Brogi, Vincenzo Calderone

**Affiliations:** Department of Pharmacy, University of Pisa, Via Bonanno 6, 56126 Pisa, Italy; ritatabanelli@gmail.com (R.T.); vincenzo.calderone@unipi.it (V.C.)

**Keywords:** opioid receptors, drug addiction, drug withdrawal, opioids, opioid use disorder (OUD)

## Abstract

This review article offers an outlook on the use of opioids as therapeutics for treating several diseases, including cancer and non-cancer pain, and focuses the analysis on the opportunity to target opioid receptors for treating opioid use disorder (OUD), drug withdrawal, and addiction. Unfortunately, as has been well established, the use of opioids presents a plethora of side effects, such as tolerance and physical and physiological dependence. Accordingly, considering the great pharmacological potential in targeting opioid receptors, the identification of opioid receptor ligands devoid of most of the adverse effects exhibited by current therapeutic agents is highly necessary. To this end, herein, we analyze some interesting molecules that could potentially be useful for treating OUD, with an in-depth analysis regarding in vivo studies and clinical trials.

## 1. Introduction

Considering the last update provided in January 2023, opioid addiction and opioid use disorder (OUD) represent a global epidemic [1]. OUD is defined by physical and/or psychological reliance on legal (prescribed drugs such as oxymorphone or hydrocodone) and illegal opioids (heroin or fentanyl, and the derivatives from the opium resin obtained from *Papaver somniferum*) [2,3]. It has been established that in the U.S., about three million citizens have had or currently suffer from OUD, while worldwide, about 16 million individuals suffer from this disorder [1,4]. The diagnosis of OUD is based on the guidelines presented in the Diagnostic and Statistical Manual of Mental Disorders-5 (DSM-5) [5,6], in which it has been reported that OUD is defined as a “problematic pattern of opioid use leading to clinically significant impairment or distress” [2]. Opioid medications are employed for the treatment of moderate-to-severe chronic cancer and non-cancer pain [7,8,9,10,11,12], although some opioids can be used for treating diarrhea [13,14,15] and cough [16,17]. Moreover, opioids can be used by patients suffering from chronic backaches [18,19] and headaches [20,21], by people recovering from surgery [22,23,24], and by both adults and children who have suffered serious injuries in falls, while playing sports, or in auto accidents, etc. [25]. Finally, opioids can be prescribed as palliative care or for limiting the suffering of patients at the end of life [26,27,28]. Despite the possible therapeutic role of opioid medications, the treatment of the above-mentioned diseases with opioids is still under debate for several reasons. In fact, in long-term therapy, the well-established phenomenon of psychological addiction occurs [29,30,31]. Furthermore, as these drugs become more widely available, there are increasing problems of abuse and diversion, which undermine the clinical efficacy of these therapeutic agents and pose a public health concern. Last but not least, the fact that these powerful analgesic agents are linked to significant adverse effects and problems has an impact on the use of opioid derivatives for treating chronic pain [29]. Accordingly, the main adverse effects of opioid prescription treatment include drowsiness, constipation, vomiting, dizziness, nausea, respiratory depression, tolerance, and, as mentioned, physical dependence. Delays in stomach emptying, hyperalgesia, hormonal and immunological alterations, muscular rigidity, and myoclonus are some less frequently observed adverse effects. Constipation and nausea are the side effects of opioid use that are most frequently reported and extremely difficult to manage [29,32,33,34,35,36]. The mentioned undesired effects could be so severe as to require the interruption of opioid use, which also increases the risk of underdosing and insufficient analgesia [37]. In particular, physical dependence and addiction are the main and most severe clinical issues that may prevent proper prescribing, and in turn, may cause insufficient pain management; they represent the most prominent outcomes of OUD [38]. In fact, people who become dependent on an opioid prescription and quit taking it may experience significant withdrawal symptoms following the last dose. For these reasons, many people encounter severe difficulties in stopping opioid use, either legally prescribed or illicit, and it can be incredibly painful (there may be severe pain in the muscles and bones, diarrhea and vomiting, cold flashes with goosebumps, involuntary movement of legs, severe problems in sleeping, severe craving, and compulsive drug-seeking behavior) [39,40,41]. Together, pain and the use of opioids have enormous societal costs. It has been estimated that pain-related healthcare and lost productivity have together cost more than $635 billion, while opioid abuse-related healthcare, criminal justice, reduced quality of life, and life lost due to overdose cost more than $1.03 trillion per year [42,43,44]. Accordingly, the next generation of opioid therapeutics is highly necessary in order to maintain the known efficacy of opioids, but engender a dramatic reduction in side effects [45]. In this review article, we analyze the main advances in the discovery of novel opioid agents, and how they can be useful in replacing the existing opioid agents, in order to treat OUD.

## 2. Opioid Receptor Functioning and Common Approved Drugs for Treating OUD

Opioids exert their pharmacological effects by targeting opioid receptors. These transmembrane proteins are members of the superfamily of class A G-protein-coupled receptors, and they are responsive toward both endogenous (such as enkephalins, dynorphins, endomorphins and endorphins) and exogenous opioids. Opioid receptors, which are found in the central nervous system and peripheral tissues, are divided into three main classes: mu (μ), kappa (κ) and delta (δ), with different subtypes (Table 1). Moreover, another member of this class, opioid receptor-like 1 (ORL1), has been identified, but there is debate about its inclusion in the opioid receptor family [46,47,48,49,50,51]. These seven-transmembrane helix group members are coupled to inhibitory Gi/o proteins. G protein (Gαi and Gβγ) signaling pathways, mitogen-activated protein kinase (MAPK), and arrestin signaling pathways are used by these receptors to transduce extracellular information [52,53,54]. In response to the significant agonist activation provided by endogenous or exogenous opioids, Gα and Gβγ subunits separate from one another to affect a number of intracellular pathways. In particular, opioid receptor activation can lead to (i) the inhibition of adenylyl cyclase, which leads to a depletion of cyclic adenosine monophosphate (cAMP); (ii) the modulation of the calcium and potassium ion channels, which modifies the concentration outward shift of K^+^, thus reducing intracellular influx of Ca^2+^; and (iii) an altered gene expression [53,55,56,57]. Considering that differential expression of opioid receptors and related endogenous ligands occurs throughout the brain, their activation could culminate in different effects [58]. For instance, the activation of opioid receptors located in the presynaptic terminals of nociceptive fibers prevents the release of the neurotransmitters implicated in pain transduction, such as glutamate, substance P, and calcitonin gene-related peptide [57]. Likewise, the activation of opioid receptors on GABAergic neurons inhibits γ-aminobutyric acid (GABA) release in the ventral tegmental area, and this event can allow dopaminergic neurons to release more dopamine into the nucleus accumbens [40,57]. Accordingly, since opioid receptors are widely expressed, they are implicated in a variety of physiological and behavioral functions, such as nociception, fear learning, social memory, drug reward and consumptive behavior, immune activation, stress and emotion, and a number of physiological functions, including gastrointestinal tract motility and respiration (Table 1) [59,60,61,62,63].

As previously mentioned, tolerance and dependence are undesired complications related to the use of legal and illegal opioids. Morphine and other opioids generate higher tolerance and dependence than any other class of therapeutic agents when used repeatedly. As a result of tolerance, higher opioid doses are necessary to provide some effects. Once the level of tolerance is high, the maximum response to the opioid is similarly diminished. The primary cause of tolerance is the functional desensitization of opioid receptors caused by their functional dissociation from specific G-proteins, which uncouple the receptors from their effector components [64,65]. In particular, specific kinases have the ability to phosphorylate opioid receptors after activation, which causes G-protein disconnection, with the consequent binding of β-arrestin. Desensitized receptors are made inactive at the plasma membrane via β-arrestin pathway signaling, which also makes it easier for them to be endocytosed and subsequently degraded or recycled. As a result, these cellular pathways play a crucial role in promoting tolerance development at various levels [66,67,68].

Although the terms dependence and tolerance are frequently considered synonyms, they indicate separate phenomena. For example, if opioid drug consumption is stopped, or an opioid receptor antagonist such as naloxone (Figure 1) is administered, dependence is canceled. The result is a withdrawal or abstinence reaction. The withdrawal response is an extremely intricate process involving numerous areas of the brain. In humans, buprenorphine-precipitated withdrawal can be observed after a single dose of morphine, because dependence develops much more quickly than tolerance. The central noradrenergic cell group known as the locus ceruleus, which is thought to play a significant role in opioid withdrawal, shows enhanced adenylate cyclase activity after chronic morphine therapy. This finding supports the long-standing theory that adenylate cyclase is involved in opioid withdrawal. However, the exact mechanism of its involvement is still unclear [69]. Furthermore, dysregulation of other neurotransmitters has been reported to be crucial in opioid withdrawal symptoms. In particular, the brain ceases making dopamine on its own after prolonged opioid usage, and becomes dependent on opioids for these effects. After becoming addicted to opioids, people who quit using them or are prescribed opioid receptor antagonists (such as naloxone) suddenly experience symptoms such as anxiety and depression, because their brains create less dopamine [69]. The administration of naloxone or another opioid antagonist is correlated with an increment in norepinephrine and dopamine release, and triggers systemic withdrawal symptoms [70,71]. Moreover, it has been demonstrated that serotonin levels also increase during opioid withdrawal [72]. Finally, other neurotransmitters, such as glutamate, are involved in opioid addiction and withdrawal. This neurotransmitter, acting on its receptors, can influence the release of dopamine in opioid addiction and withdrawal. Accordingly, the activation of the glutamatergic system in particular brain areas (i.e., nucleus accumbens) during morphine withdrawal may be crucial regarding negative effects [73]. Acute opioid withdrawal causes stress and activates the hypothalamic–pituitary–adrenal (HPA) axis, increasing the release of adrenal cortisol, adrenocorticotropic hormone (ACTH), and pituitary pro-opiomelanocortin mRNA [74,75,76].

Dependence is one of the main hallmarks of OUD that is characterized by physical and physiological dependence on prescription and/or illicit opioids. Most patients suffering from OUD and individuals with abstinence induced by medically supervised withdrawal may require long-term care to avoid relapse. The most common first-line treatment for people with OUD consists of the administration of an opioid receptor agonist or antagonist, as well as adjuvant psychosocial counseling. Buprenorphine, methadone, and naltrexone (Figure 1) are FDA-approved drugs for treating OUD, whereas naloxone is not considered a therapeutic agent for the treatment of OUD per se, being approved by the FDA for the diagnosis or treatment of the respiratory depressive symptoms of opioid use that can lead to fatal opioid overdose. Furthermore, when promptly delivered, this medication has been shown to decrease the morbidity and mortality related to opioid overdoses [77].

Buprenorphine is a ligand of opioid receptors with relevant affinity. It behaves as an antagonist of the δ- and κ-opioid receptors, and as a partial agonist of the µ-opioid receptor and opioid receptor-like 1 (Table 1). Buprenorphine is just a partial agonist, so it cannot completely replace other opioids (heroin, codeine, and oxycodone) acting on the µ-opioid receptors. Similar to methadone, buprenorphine can ease a patient going through opioid withdrawal [78]. Notably, if the patient uses opioid derivatives while taking buprenorphine, the rewarding effect is significantly reduced due to its partial agonist effect. Taking advantage of this effect, buprenorphine shows a limited risk of overdose with respect to methadone and other opioids, and has a limited impact on respiratory depression [79,80]. Currently, buprenorphine is available alone or in combination with the opioid receptor antagonist naloxone. Although some studies indicate that patients who taper off buprenorphine experience higher rates of relapse than those maintained on the medication for a longer period, both formulations of buprenorphine are still helpful for treating OUD [81]. In a Swedish trial, patients who received 16 mg/day of buprenorphine were compared to a control group who received the drug for detoxification for 6 days before receiving a placebo. Psychosocial support was provided to all patients. In this trial, buprenorphine showed a treatment failure rate of 25% compared to 100% for placebo. Treatment discontinuation occurred when there were more than two opioid-positive urine tests within three months; hence, relapse was directly associated with treatment retention. Notably, a 20% death rate was detected for patients who were not maintained with treatment [82]. According to a meta-analysis, buprenorphine diminished the amount of opioid-positive drug tests by 14.2%, and increased the likelihood of patients staying in treatment by 1.82 times compared to placebo-treated patients [83]. Accordingly, buprenorphine must be administered at a dose high enough to be effective (typically, 16 mg/day or more). Buprenorphine treatment failure and the false belief that the drug is not effective result from certain treatment providers who were reluctant to use opioids prescribing lower doses for shorter treatment durations [84]. Interestingly, both methadone and buprenorphine work equally well in cutting down opioid consumption. There were no differences in opioid-positive drug tests or self-reported heroin usage while patients were treated with methadone or buprenorphine at medium-to-high doses, according to a thorough review comparing methadone, buprenorphine, and placebo. Buprenorphine doses of 6 mg or less and flexible dosing regimens are particularly ineffective in keeping patients in treatment compared to methadone, emphasizing the importance of using evidence-based dosing regimens for these drugs [83].

Methadone is a synthetic, long-acting opioid agonist. The µ-opioid receptors in the brain are totally activated by methadone in the same way that they are by prescription and illegal opioids. In therapeutic doses, in people suffering from OUD, methadone reduces the euphoric “highs” of shorter-acting opioids (heroin, oxycodone, and codeine) and the unpleasant “lows” of opioid withdrawal. Nevertheless, it may take days to weeks to obtain a therapeutic dose, and this dose must be customized for each patient to limit cravings and discourage the further use of opioids [85,86]. Interestingly, the effectiveness of methadone treatment and psychosocial support in comparison with placebo was detailed in a recent study. The research proved beyond doubt that methadone was successful in lowering opioid usage, the spread of infectious diseases linked to opioid use, and criminality. Methadone users, compared to controls, had 33% fewer opioid-positive drug tests and were 4.44 times more likely to complete their treatment. Long-term (more than six months) outcomes were better in groups receiving methadone, independent of the frequency of counseling received. Methadone treatment considerably improves outcomes, even when administered in the absence of frequent counseling sessions [87,88].

Naltrexone is a full antagonist of the µ-opioid receptor. Its pharmacological behavior is relevant in preventing the effects of all opioid derivatives, such as euphoria and analgesia [78]. Naltrexone does not result in physical dependence or any of the pleasurable benefits of opioids. In theory, extended-release naltrexone patients quickly learn to abstain from the opioids that led to their addictive behaviors, and following regular usage of the drug, their cravings subside [89,90]. Naltrexone, formulated for oral administration, has received approval for treating OUD. Although the drug does not produce tolerance or withdrawal, the efficacy of this formulation has been principally constrained by poor treatment adherence [91]. Based on the available research, the effectiveness of oral naltrexone as a treatment for OUD is therefore not sufficiently demonstrated [92]. On the contrary, there is no need for daily dosing with extended-release injectable naltrexone, because it is given monthly. Even though this formulation is the most recent type of OUD treatment, evidence to date indicates its effectiveness [93].

The double-blind, placebo-controlled experiment that had the biggest impact on the FDA’s decision to approve naltrexone for treating OUD in 2010 also demonstrated that the drug significantly boosted opioid abstinence. Briefly, 90% of the confirmed abstinence weeks were in the naltrexone group, versus 35% in the placebo group. The naltrexone-treated group also had a greater rate of treatment completion (58% vs. 42%), as well as lower rates of subjective drug craving and relapse (0.8 vs. 13.7%). The naltrexone group continued to improve over the course of a 76-week open label period. To determine its effectiveness in a larger population, more research is necessary [89,94,95].

According to a National Institute on Drug Abuse (NIDA) clinical trial, extended-release naltrexone and a buprenorphine/naloxone combination showed equal effectiveness in treating OUD after treatment began. Starting therapy with naltrexone among active opioid users was more challenging, because this opioid requires total detoxification. The naltrexone formulation, however, was just as effective as the buprenorphine/naloxone combination once detoxification was complete [96].

Buprenorphine, like methadone, maintains opioid tolerance and physical dependence in patients; thus, discontinuing its use can result in withdrawal, even if buprenorphine withdrawal symptoms can be milder. The beginning of non-life-threatening opioid withdrawal following the first dose of buprenorphine is the most significant side effect for OUD patients. Furthermore, people with OUD who start buprenorphine treatment immediately have a lower risk of passing away from an opioid overdose [97]. Notably, complications related to the use of buprenorphine, such as hypogonadotropic effects, QT prolongation, or cardiac arrhythmias, are limited with respect to the use of methadone [98]. It is crucial to remember that since methadone and buprenorphine are opioids, overuse is possible. Buprenorphine and methadone, like other opioids, can cause physical dependence and a diagnosable OUD, necessitating their secure storage and exclusive use by the person to whom they are prescribed.

It is significant to remember that first-line therapy for OUD is not devoid of possible undesired effects due to drug–drug interaction issue. Recently, in an interesting work, Berger and colleagues reported a retrospective study in which the potential for drug–drug interaction in patients with OUD was assessed. Starting from the fact that some treatments for OUD show common metabolic pathways, the use of these drugs could be related to the insurgence of drug–drug interactions. For example, buprenorphine and methadone are metabolized by CYP3A4, so other drugs able to both inhibit or induce CYP3A4 enzymes could be relevant in drug–drug interactions, and could modify the duration and intensity of their effects. In addition, some of the drugs used for treating OUD, such as methadone, show an increased risk of QT prolongation, so the use of other drugs with similar effects could act synergistically, leading to fatal arrhythmias.

In fact, in the aforementioned work, it was reported that QT prolongation (24.2% inpatient, 45.8% discharge) and additive central nervous system effects/respiratory depression (68.8% inpatient, 50.6% discharge) were the two most prevalent categories of drug–drug interactions. Instead of being considered contraindicated, many drug–drug interactions were labeled as requiring strict monitoring. Opioids, benzodiazepines, antipsychotics, and anti-infective agents were the four drug classes with a risk of drug–drug interactions in the inpatient setting. Furthermore, due to the increase in the use of drugs to treat OUD, specialists (i.e., physicians, pharmacists) will require the ability to identify and manage potential drug–drug interactions [99].

Considering that the use of the aforementioned drugs is associated with an increase in undesired side effects, to exploit their pharmacological potential, it is necessary to identify novel opioid receptor ligands devoid of most of the adverse effects exhibited by current therapeutic agents. To this end, in this review article, in the following sections, we introduce some interesting molecules that could be potentially useful for treating OUD, including their pharmacological activities observed in in vivo studies and in clinical trials.

## 3. Psychosocial Interventions for Substance Abuse and Dependence

Although medications are powerful tools to prevent morbidity and mortality related to OUD, the effectiveness of these treatments is limited by problems at all levels of the care cascade, including diagnosis, entry into treatment, and retention in treatment. Medically managed withdrawal is typically insufficient to produce long-term recovery, and may increase the risk of overdose in individuals who have lost their tolerance to opioids and resume using them [100,101]. Additionally, oral opioid agonist therapy (OAT) is only accessible through accredited programs for treating addiction or from physicians who have completed specialized training in opioid medicine. Access to medication for OUD in primary care and specialty settings (pain and infectious diseases clinics, psychiatrists, and emergency departments) still faces misconceptions about the medications themselves and their use. Restrictions on who can prescribe them, and stigma toward methadone and buprenorphine are constantly observed, considering that there is a perception that one addictive drug is replaced by another [100,101,102]. Linkage and retention in care are actually challenging due to the lack of acceptance of OUD as a medical disease and the lack of willingness among many clinicians to treat OUD with medications. This is because patients and doctors may view those who have addiction as manipulative and unworthy of care [101]. As a consequence, some patients with OUD do not obtain access care, some others cannot be included in them, patients might stop responding after a period of benefit, and many patients are unable to adhere for sufficiently long periods to medications for treating OUD, since buprenorphine and methadone are frequently administered when they are supplied at excessively low doses or for insufficiently long periods of time [101,102]. Notably, despite medications significantly improving OUD outcomes, a relevant number of patients continue the use of illegal drugs during OAT. This population of patients, namely intermittent responders (who stop or significantly diminish their drug use in the first weeks, but later relapse and cycle through periods of unapproved exit and re-admission) and brief responders (who only achieve brief periods of reduced drug use), and poor responders (who do not significantly diminish their use of illegal drugs) could benefit from psychosocial interventions [103].

The first step toward an improved management and consequent treatment of OUD must consider a better identification and diagnosis of the disorder, which means that practitioners should be educated throughout healthcare to screen and treat OUD, since OUD only manifests as opioid withdrawal in a few cases, while in most cases, it does not present with any acute symptoms. It might be beneficial to identify opioid misuse or OUD through screening instruments that could warrant a more in-depth assessment of the severity of the disease, although most do not specify the type of drug used. At this stage, the role of clinicians becomes fundamental; avoiding risk of embarrassment or stigmatization is pivotal for facilitating a serious discussion on drug abuse, as well as the referral of patients for appropriate care when not equipped to provide adequate treatment [101]. Second, incorporating non-pharmacological interventions as part of the treatment program could implement adherence to therapy and prevent relapses, and reach those hard-to-reach patients who struggle to respond to therapy. Among these psychosocial treatments, motivational interventions include working with patients to encourage motivation to change, enhancing adherence to therapy through education, and keeping motivation high, which is essential for a positive outcome and the likelihood of treatment adherence [100,104]. Cognitive behavioral treatment (CBT) has been demonstrated to be successful in preventing relapses [104,105]. The adoption of appropriate coping mechanisms to prevent substance use in high-risk settings is encouraged via CBT, which teaches patients how to recognize high-risk situations and triggers [106]. Other interventions with a reinforcement focus, such as contingency management (CM), which entails the administration of rewards or vouchers given to patients in exchange for demonstrating a desired behavior such as abstinence or adherence to a medication, may also increase their compliance with substance use disorder treatment [104,106]. CM increased the number of drug-negative urinalysis screens compared to a control group, and this effect was stronger for immediate rewards, per a meta-analysis that examined the use of CM to promote opioid abstinence in patients receiving methadone; however, larger effect sizes appear to be associated with a shorter treatment duration, indicating that the effects of CM could wane over time [107,108]. In fact, evidence indicates that CM’s reward effects diminish, whereas CBT strategies gain momentum over time [109,110]. According to a study, methadone maintenance therapy patients who also received combination CBT and contingency management experienced a significant 12-week reduction in craving [111]. Relapse prevention (RP) focuses on recognizing and responding to situations or events that are more likely to lead to drug use by assisting people in avoiding or managing them by practicing alternate (non-drug) reactions [104,112]. RP methods comprise challenging the patient’s perception of the benefits of usage, and offering psychoeducation to assist the patient to make correct decisions in potentially dangerous circumstances [104]. In a meta-analysis, scientists showed the effect sizes of different types of psychosocial treatments, as well as abstinence and treatment retention rates for cannabis, cocaine, opioids, and polysubstance abuse and dependence treatment trials. The findings indicated that relapse prevention and other CBT came in second place, with the largest effect size estimations [112]. When entering addiction treatment, the specific opioid withdrawal and initiation of buprenorphine–naloxone issues should be considered. In fact, a method that helps people manage their opioid addiction and the unpleasantness of opioid withdrawal could be beneficial to them. With the help of mindfulness-based therapies, patients can learn to observe their emotional and physical states without reacting, giving them the freedom to act in ways that are more consistent with their values, and to deal with cravings and urges, in order to stop using drugs [105,113,114,115,116]. Twelve-step facilitation (TSF) therapies are a collection of semi-structured treatments designed to assist people to stop using alcohol and other drugs by meticulously fostering links with and encouraging active engagement in neighborhood 12-step mutual help groups. In a parallel-group, randomized clinical trial, ten sessions of either motivational enhancement therapy/CBT or a novel integrated TSF were compared. The latter was able to increase 12-step community mutual help organization participation among adolescent outpatients, and to produce lower substance-related consequences both during and after intervention [117].

In addition to psychosocial interventions, prevention strategies, when implemented in childhood and adolescence, reduce later drug use, which includes prescription opioid abuse. These interventions would affect every type of opioid abuse. Although primary or secondary prevention intervention is a high priority and a key feature of a comprehensive approach to treating OUD, fewer efforts have been made in this regard compared to efforts to enhance the prescription of opioid analgesics and expand the availability of naloxone to prevent overdoses. These approaches entail adopting evidence-based substance use disorder preventive treatments in family, educational, and/or social contexts [102], as well as patients’ education about the disease of addiction, its treatment, and overdose risk, identification, and response [101]. Accordingly, information on naloxone, the availability of medically supervised injecting rooms, and community services for addiction treatment should all be included in overdose prevention education. To prevent invasive infections associated with injecting drugs, safer injecting practices such as injection site preparation, the use of filters and clean water, and the use of sterile non-reused needles and syringes should be considered [118]. Drug-checking services (DCS) and harm reduction strategies mandated by law give drug users the chance to have their substances tested before consumption, and to consume previously acquired narcotics under the supervision of medical specialists [119,120]. Based on data from multiple studies, it has been determined that supervised injection facilities are primarily linked to significant decreases in opioid overdose morbidity and mortality, improvements in injection practices and harm reduction, significant increases in access to addiction treatment programs, positive behavior change, and the reduction of harm without having any impact on crime or public nuisance [119,121,122,123,124,125,126].

Overall, opioid addiction can be successfully treated with both opioid agonists and antagonist drugs. However, when given as a part of a CBT, pharmaceutical treatment is most successful. In fact, the integration of education, motivational enhancing techniques, and self-help groups into individual and group counseling approaches in inpatient and outpatient programs helps patients to change their perspectives on how opioids affect their lives and to realize that change is possible; this works to reduce behaviors that support illicit drug use while developing new behaviors that reduce drug-related problems. Last but not least, an integrated approach allows us to tailor therapy to patients’ needs, to optimize clinical management and improve overall outcomes.

## 4. Mechanisms of Drug Withdrawal and Novel Therapeutic Options for Treating OUD

Due to the need to restrain the OUD crisis, which is a public health issue worldwide, research has focused on finding novel drugs, either synthetic or naturally derived, that can effectively treat opioid addiction, avoiding the limitations of drugs used for treating OUD [127,128]. Indeed, existing treatments for OUD, namely methadone, buprenorphine, and naloxone, target the disease at the opioid receptor: ORT, also known as OAT or opioid substitution therapy (OST), reduces overdoses and the damaging effects of substance abuse, although it does maintain tolerance and physical dependence on opioids [102,129].

Despite the fact that the opioid crisis was caused by overprescription of opioid drugs, research into new treatments for OUD takes into account, on one hand, the discovery of partial agonists or mixed agonist/antagonists which work through the same receptors; this could allow for antinociceptive effects during the withdrawal process and OUD treatment, thereby avoiding some of the main complications related to the opioid treatment. On the other hand, designing and synthesizing innovative antinociceptive agents with reduced undesired effects may be pivotal in resolving and preventing the opioid crisis [127,128,130,131,132].

Before deepening our analysis of the novel strategies of interaction concerning opioid receptors, it is worth mentioning that although it is not the focus of this article, much of the research carried out in the last few years focuses on looking beyond the opioid receptor [127,133,134]. Indeed, progress in neuroimaging has allowed for the discovery of possible therapeutic targets in addiction, including the prefrontal cortical reward network, as well as the development of interventional psychiatric therapeutic strategies that act via the modulation of the higher order neural circuitry involved in behavior [127]. Transcranial magnetic stimulation (TMS) [135], transcranial direct current stimulation (tDCS) [136], deep brain stimulation (DBS) [137] (NCT04354077; NCT03950492), vagus nerve stimulation (VNS) [138], and focused ultrasound (FUS) [139] are potential interventional treatments for OUD, currently FDA-approved for treating depression, anxiety, epilepsy, Parkinson’s disease, tremor, alcohol and cigarette use, and cocaine or methamphetamine use [140]. The potential benefit of these treatments is that they do not continue to rely on opioids and can be used as a stand-alone, non-drug treatment for OUD or as an additional treatment to the currently existing OAT. However, they are invasive because they require the implantation of a device in the brain or neck, which carries risks associated with surgery. Moreover, since some of these interventions are presently FDA-approved for treating resistant anxiety and depression, which are often linked to OUD, the treatment of these comorbidities with interventional approaches could improve medication experience and adherence [140].

The therapy of depression, anxiety, and addiction to tobacco and alcohol is also based on preclinical evidence of psychedelic agents. Serotonergic hallucinogens such as lysergic acid diethylamide (LSD) act as agonists of the 5-HT_2A_ receptor, specifically those located on the dendrites of cortical pyramidal cells, while at the anatomic level, they increase structural neuroplasticity and the synthesis of neurites, dendritic spines, and synapses in the brain areas related to emotion processing and social cognition [141,142,143]. These data suggest a possible use of psychedelic agents in substance abuse, because several of these drugs are not related to physical dependence and are usually tolerated at significantly high doses with no associated permanent adverse effects [127].

Further possible innovative therapeutic strategies indicated for treating OUD comprise genetic and gene product approaches. According to genetic epidemiology research, genes account for around 50% of the risk of developing substance use disorders, including OUD. Even though no particular genes have been found that could be used as OUD biomarkers, a series of genes appear related to the cause of opioid addiction. Among them are the gene that encodes the μ-opioid receptor, *OPRM1*, the gene that modulates the trafficking and gating properties of AMPA (α-amino-3-hydroxy-5-methyl-4-isoxazole-propionic acid) receptors, *CNIH3*, the gene that encodes a voltage-gated potassium channel, *KCNJ6*, the dopamine receptor D2 gene (*DRD2*), and the brain-derived neurotrophic factor (*BDNF*) gene [101]. Alongside the classic genetic research, the epigenetic approach is gaining attention. In response to both internal and external stimuli, hundreds of genes can have their expression influenced by microRNAs (miRNAs), which are pleiotropic epigenetic regulators of gene expression. In fact, it has been established that numerous miRNAs could play a crucial role in synaptic plasticity and drug addiction. Neuron-specific miRNAs have been detected in the bloodstream in response to at least one drug of abuse (tobacco), implying that these could represent biomarkers for addiction and potential drug targets for developing innovative therapeutics [144]. One of the latest tools that is under investigation for preventing and treating opioid addiction is gene editing, with novel clustered regularly interspaced short palindromic repeats (CRISPR) [134].

Another possible therapeutic strategy to treat OUD is immunotherapy. This approach has been demonstrated to improve adherence and reduce the risk of relapse. Among its advantages, it is necessary to mention the use of selective drugs of abuse, which is unlike the use of the non-selective orally administered μ-opioid receptor antagonists currently available. Unlike orally administered μ-opioid receptor antagonists, vaccines for the treatment of OUD do not require prior detoxification or monitoring of treatment compliance [145]. On the other hand, one of the limitations of immunotherapies for substance use disorders is that antibodies do not improve drug cravings, and can precipitate withdrawal symptoms [145,146]. Finally, a crucial feature of the design process is finding haptens with significant structural congruence with the target drug; failures to meet this requirement are likely to blame for failure of first-generation conjugate vaccines against cocaine and nicotine in clinical trials [145,146]. Meanwhile, the second generation of vaccines demonstrated better outcomes in preclinical settings, and we expect corresponding success in clinical studies. Although some encouraging results have been obtained, additional studies are necessary to obtain clinically useful products [127,145,146].

One last mention must be made regarding the use of cannabis or Δ-9-tetrahydrocannabinol (THC) as a possible therapeutic approach to alleviate opioid withdrawal. Despite preliminary evidence in this sense, further clinical studies are necessary to establish the real risk/benefit ratio of this possible therapeutic approach [133].

Lastly, public health measures and behavioral interventions at the individual level are fundamental strategies to treat OUD [101,127].

### 4.1. Promising Opioid Receptor Modulators for the Treatment and Prevention of OUD

#### 4.1.1. Methocinnamox (MCAM)

Methocinnamox, chemically 14β-(4′-methylcinnamoylamido)-7,8-dihydro-*N*-cyclopropylmethyl-normorphinone, (MCAM), is a potent, long-lasting, pseudo-irreversible μ-opioid receptor antagonist [130,147,148]. First described by Broadbear and colleagues [149], MCAM is structurally similar to buprenorphine; it reversibly binds μ- and δ-opioid receptors without interacting with other nociceptors [130,147,148]. The mechanism of action of MCAM was well described by Zamora and colleagues, who tested the time-dependent nature of the antagonism, including its non-surmountability by agonists and its lack of reversibility binding at μ-opioid receptors, both in vitro and in vivo [147]. Indeed, in HEK cells expressing the human μ-opioid receptor, MCAM pretreatment diminished the maximal response to the μ-opioid receptor agonist, DAMGO ([D-Ala2, *N*-MePhe4, Gly-ol5]-enkephalin), in a time-dependent and non-washable manner. Similarly, this effect was shared by the irreversible antagonist, β-FNA (β-funaltrexamine). In contrast, the competitive antagonist naloxone was completely surmountable by DAMGO, and its antagonism was not dependent on time and totally reversible upon washout. Conversely, the binding of MCAM was found to be reversible or pseudo-irreversible. The idea that MCAM may have both a (pseudo)irreversible orthosteric action to diminish the agonist’s maximal response and an allosteric action to change the µ-opioid receptor agonist’s potency offers one explanation for the slow off-rate of MCAM from the orthosteric location. Interestingly, pretreatment of cells with naloxone, at a concentration to completely occupy the μ-opioid receptor orthosteric site, before the MCAM administration, followed by a significant washing process for removing naloxone and the unbound MCAM, prevented the MCAM-induced reduction in the maximal response to DAMGO but did not block the dextral shift in the DAMGO concentration–response curve. This result is consistent with the presence of a naloxone-insensitive (allosteric) site targeted by MCAM that could mediate a reduction in the potency of DAMGO, as demonstrated by the shift in the DAMGO concentration–response curve and by a differential modulation of opioid agonist responses. The pseudo-irreversible binding incapacitated the μ-opioid receptors; thus, cells should synthesize nascent μ-opioid receptors to reestablish previous functionality. For this reason, MCAM showed an exclusively long duration of action (DOA) [130].

Additionally, in cells expressing the human δ- or κ-opioid receptors, pretreatment with MCAM shifted the concentration–response curves to the δ-agonist, DPDPE (D-Pen2,5]-enkephalin), and the κ-agonist, U50488, to the right in a surmountable, time-independent and fully washable manner, suggesting that MCAM could act as a reversible competitive antagonist in δ- and k-opioid receptors [147].

Although not tested in human trials yet, data obtained from animal studies showed a medication that could exert an important positive influence on the treatment of opioid overdose as well as OUD [150,151,152,153]. As a matter of fact, the administration of MCAM (0.32 mg/kg) to Rhesus monkeys before and after injection of heroin prevents and reverses respiratory depression in a similar manner to naltrexone (0.032 mg/kg) and naloxone (0.0032–0.1 mg/kg), respectively. However, the persistent effects of MCAM represent the most impressive difference; in the prevention study, MCAM could attenuate the respiratory depressant effects of heroin for at least 4 days, whereas the antagonist effects of naltrexone disappeared in 1 day. Likewise, in the reversal study, MCAM protects against the respiratory depressant effects of heroin on the day of administration as well as on subsequent days when additional doses of heroin are given, according to the heroin dose–effect curve that did not return fully to control values within 8 days; however, the administration of the largest dose of naloxone caused a return to baseline values within 45 min [152]. Additionally, the impact of MCAM on the abuse-related effects of opioids was characterized by examining its capability to attenuate opioid self-administration in Rhesus monkeys. In one experiment, intravenous infusions of heroin (0.0032 mg/kg) or cocaine (0.032 mg/kg) were delivered according to a fixed-ratio schedule of reinforcement; in a second trial, monkeys were given the option of eating or receiving 3.2 mg/kg of the µ-opioid receptor agonist remifentanil intravenously. In a third trial, the direct effects of MCAM (0.32 mg/kg) were studied by monitoring responses to food and physiologic variables (heart rate, blood pressure, temperature, and activity). In the heroin self-administration experiment, naltrexone (0.032 mg/kg) as well as MCAM dose-dependently decreased infusions obtained on the day of treatment; however, in naltrexone-treated monkeys, the reduction in opioid intake was significant on the day of treatment and response, returning to baseline levels the next day, while MCAM significantly decreased heroin consumption on the first day of therapy and for several days subsequently, with decreases persisting, on average, for 10 days. Neither naltrexone nor MCAM could alter the response sustained by cocaine on the day of treatment or for several days thereafter, establishing the selectivity of MCAM for attenuating opioid-maintained behavior. In monkeys responding to a food/drug choice procedure, the choice of food decreased as the dose of remifentanil increased (0.32 and 1.0 μg/kg). The injection of 0.032 mg/kg naltrexone immediately before, but not the same dose of naltrexone administered 24 h before, decreased the choice of remifentanil and increased the choice of food, shifting the remifentanil dose–effect curve rightward, before it returned to control levels the next day. The remifentanil dose–effect curve was shifted rightward and downward in a dose-dependent manner by MCAM, and the time it took for the curve to recover was inversely proportional to the dose of MCAM, with the curve fully recovering 4 days after 0.32 mg/kg MCAM administration, 8 days after 1.0 mg/kg, and 12 days after 3.2 mg/kg. The last group of monkeys were treated with MCAM 1 h before the activity session, and doses of MCAM that relevantly reduced opioid intake did not alter the rate of response, the number of pellets earned, or physiological measures such as blood pressure, heart rate, and body temperature, indicating a favorable safety profile of MCAM for these physiological parameters [153].

The same research team also evaluated the effects of acute and repeated MCAM administration on the self-administration of fentanyl, a μ-opioid receptor agonist [150]. Cocaine (0.032 mg/kg/infusion) or fentanyl (0.00032 mg/kg/infusion) were delivered intravenously to four Rhesus monkeys, and MCAM (0.1–0.32 mg/kg) or the opioid receptor antagonist naltrexone (0.001–0.032 mg/kg) were injected before test sessions to assess acute effects. To test the efficacy of repeated therapy, 0.32 mg/kg MCAM was injected once every 12 days for a total of five injections. Following acute injection, MCAM and naltrexone both reduced the amount of fentanyl that patients self-administered on the treatment day. This attenuation persisted about 2 weeks after the higher MCAM dose and for only one day after naltrexone. Repeated MCAM administration diminished cocaine self-administration but did not affect fentanyl self-administration for more than two months [150].

MCAM’s ability to revert ventilatory depression induced by fentanyl was also evaluated considering the route of administration (intravenous and subcutaneous) and compared to naloxone, the only FDA-approved medication available to treat opioid overdose. MCAM and naloxone were able to revert the ventilatory depressant effects of 0.178 mg/kg fentanyl injected i.v. in male Sprague Dawley rats, in a dose-dependent manner. MCAM, but not naloxone, decreased the ventilatory depressant effects of fentanyl the day following antagonist delivery. After intravenous and subcutaneous treatment, the duration of the effect of MCAM extended up to 3 days and at least 2 weeks, respectively. Additionally, MCAM reduced fentanyl’s antinociceptive effects, with antagonism persisting for up to 5 days and more than 2 weeks following intravenous and subcutaneous treatment, respectively [151].

Taken together, these findings emphasize MCAM’s long-lasting effects, which are likely due to its pseudo-irreversible binding to µ-opioid receptors, its capacity to safely and effectively reduce opioid self-administration for extended periods of time after a single administration, and its ability to remain effective with repeated administration, even at very low plasma concentrations. These findings support the idea that pharmacodynamic factors are important in its long-lasting effects, the quick recovery from opioid-induced ventilatory depression, and the subsequent protection from respiratory depression brought on by the re-emergence of effects of an agonist with an action duration longer than naloxone. Accordingly, MCAM may be a medication recommended for those who co-use alcohol or benzodiazepines, because it has a positive safety profile and may not cause any adverse drug reactions [150,153]. By increasing the effectiveness of the treatment and patient compliance, a drug that addresses all these issues could have a substantial positive influence on the treatment of opioid overdose and OUD. Although MCAM is being researched for its potential as a long-term OUD treatment for the opioid crisis, it is relevant to note that no testing has been performed on humans.

#### 4.1.2. Mitragynine (Kratom)

Given its wild availability and fast spread worldwide for both recreational and medicinal purposes, we believe that the *Mitragyna speciosa*, a Southeast Asian evergreen tree in the coffee family commonly known as kratom, deserved to be mentioned in this review, notwithstanding the deficiency of human data available.

According to several anonymous online surveys, kratom is primarily used by white, middle-aged Americans for several health-related purposes, such as pain relief, reduced fatigue, increased energy and focus, reduction of anxiety and depression, and as an alternative to alcohol, opioids, and/or other drugs to manage withdrawal and maintain abstinence [154,155,156].

Because of its dose-related mild stimulant- and opioid-like effects [154,156], its use is raising concerns with regulatory agencies, and has resulted in scheduling actions in a number of countries [157,158]. In fact, kratom contains more than 30 different indole alkaloids, including its main alkaloid mitragynine, which accounts for roughly 66% of all the alkaloids found in kratom leaves. Among the minor alkaloids, its naturally oxidized derivative 7-hydroxymitragynine (7-OH) is of particular interest [128,159,160,161]. In particular, these alkaloids represent a novel class of opioid receptor modulators with distinctive pharmacological characteristics: mitragynine and its oxidized analog, 7-OH, are partial agonists of the human μ-opioid receptor and competitive antagonists at the κ- and δ-opioid receptors, respectively. Additionally, 7-OH and mitragynine are G-protein-biased agonists of μ-opioid receptors that do not recruit β-arrestin after receptor activation, which has been linked to many undesirable effects of opioids, including constipation, respiratory depression, and dependence [128,161]. As a result, mitragynine did not lead to dependence or increased self-administration in animal models, and it even decreased the preceding administration of morphine. However, the results of the study presented with a dependence liability of 7-OH, which is readily self-administered, and prior exposure increased subsequent morphine intake, which is ascribable to its reinforcing effects, mediated in part by μ- and δ-opioid receptors. Pretreatment with nalxonaxine, a μ-opioid receptor antagonist, and naltrindole, a δ-opioid receptor antagonist, on 7-OH and morphine revealed a reduction in 7-OH self-administration, whereas only nalxonaxine decreased morphine intake. The results showed that 7-OH should be regarded as a kratom component with strong abuse potential that could also raise the intake of other opioids, while mitragynine does not have abuse potential and decreases morphine consumption [162]. Notably, Kruegel and colleagues highlighted that since 7-OH could not be present in all plant extracts, its potential contribution to the actions of *Mitragyna speciosa* is not universal, unless 7-OH is produced as a metabolite. Although they attributed competitive antagonist activity at µ-opioid receptors to a variety of different main alkaloids isolated from the plant, this results in a complicated interplay of antagonist and agonist actions that compete at the opioid receptors, which then translate into the strong psychoactive effects of the crude plant material [161]. In order to fully analyze the abuse potential of mitragynine, rats trained to self-administer methamphetamine had their own self-administration assessed and compared to that of heroin. Unlike heroin, mitragynine did not continue to provide reaction rates that were higher than those obtained using saline injections. While having little influence on the rates of response sustained by methamphetamine across the same range of mitragynine doses, mitragynine dose-dependently reduced the rates of response maintained by heroin. These findings collectively imply a low risk of abuse of mitragynine and the possibility of using mitragynine therapy to minimize opioid abuse in particular [163]. To deepen the characterization of kratom alkaloids, using a radiant heat tail flick assay on mice, the antinociceptive profile of mitragynine and its derivatives was assessed. Mitragynine induced antinociception that was 66 times less strong than morphine following subcutaneous injection, although 7-OH was around five times more potent [128]. Of particular interest, mitragynine pseudoindoxyl, a rearrangement product of 7-OH with a spiro-pseudoindoxyl nucleus, was found to be 1.5-fold more potent than morphine after intracerebroventricular administration and 3-fold more potent following subcutaneous administration; it showed a shorter duration of antinociceptive effect than morphine, with a peak effect at 15 min, and also displayed activity following oral administration [128].

Since only one study was conducted in Thailand in 2015, and no other controlled clinical trials using kratom or its alkaloids have been undertaken, the behavioral pharmacology and abuse potential of kratom have not been fully described in a large human population to date [155]. The mentioned prospective study enrolled ten chronic, regular, healthy kratom consumers. By administering a known dose of kratom tea (60 mL) daily for seven days before the experiment, the steady state was modified in each subject. On day 8, a loading dose of mitragynine tea was administered, and blood concentrations of the drug were assessed at 17 time points. Additionally, urine concentrations during the 24 h period were collected and quantified using the liquid chromatography–tandem mass spectrometry technique. According to the outcomes, the authors proposed linear pharmacokinetics, which followed an oral two-compartment model, hypothesizing a hepatic metabolism. No severe undesired effects were detected by kratom users during the trial, while some mild events included tongue numbness after having finished drinking kratom tea, and an increment in blood pressure and heart rate, although the onset occurred 8 h after consuming kratom tea [164].

Given the lack of controlled human studies, surveys represent one effective way, although not fully reliable, to provide initial insights and understand more about outcomes following kratom exposure, and to inform prospective evaluations of kratom for diverse indications as well as regulatory decisions regarding scheduling. Data collected from surveys conducted in the U.S. mostly agree that the primary reasons for kratom use include increased energy and focus, alleviating pain, anxiety, or depression, cutting down on or quitting the use of prescription opioids or heroin, and relieving withdrawal symptoms. Respondents evidenced dose-dependent detrimental effects including constipation, nausea, drowsiness, and dizziness as the most commonly detected adverse reactions, generally perceived as mild and less than 24 h in duration [154,155,156,165]. The self-selected convenience sample is the main drawback of internet-based surveys of kratom users, because it is likely to exhibit selection bias in favor of people who are younger and more likely to be positive about kratom, which understates the use of the preparation by older participants [155,156].

In conclusion, mitragynine (kratom), one of the main alkaloids of *Mitragyna speciosa*, deserves further research, not only as a possible pharmacotherapy for OUD with low abuse potential, but also as an effective new medication with minimal abuse liability to manage acute and chronic pain, especially due to its public health impact. Since history and its patterns of use and effects confirm its benefits to consumers, some authors suggest that making kratom illegal could lead to greater opioid consumption among those who are now using it to stop taking opioids, and to the establishment of a black market [154,165].

### 4.2. Safer and More Tolerable Therapeutics for Treating Severe Pain: Oliceridine, the Opioid of the 21st Century?

As previously mentioned, the misuse and diversion of opioid analgesics were made easier by their overprescription, and patients with chronic pain were put at risk of addiction and overdose without necessarily experiencing any relief from their agony. Postoperative pain treatment continues to be a prevalent difficulty in contemporary medicine, despite the implementation of prescription drug monitoring programs in all states, alongside better physician education in opioid prescriptions and pain management aiming to prevent inappropriate dosing [102,166]. As a result, the development of new and safer pain medications that warrant effective pain management is needed.

Oliceridine (formerly known as TRV130, C_22_H_30_N_2_O_2_S·C_4_H_4_O_4_) represents a novel class of opioid medicines able to target μ-opioid receptors acting as biased ligands. Opioid analgesics are full agonists at the µ-opioid receptors, which after receptor activation, engage two independent transduction pathways (the G-protein-coupled signaling and the β-arrestin pathways) with totally different pharmacological effects. G-protein signaling is predominantly involved in analgesia, reward, and liking, whereas the β-arrestin pathway is implicated in negative effects such as respiratory depression and gastrointestinal symptoms, as well as the attenuation of analgesic benefits [166,167,168,169,170,171]. By downregulating the β-arrestin pathway and selectively activating the G-protein-coupled signaling pathway, olceridine has the potential to limit the occurrence of opioid-related adverse events and increase the therapeutic window, thereby satisfying the need for an analgesic with the effectiveness of a conventional opioid but with a more manageable side effect profile [166,167,168,172].

Oliceridine formulated as a fumarate salt [166] was first discussed for potential approval at the FDA’s meeting regarding anesthetic and analgesic drug products in October 2018 [173]; recently, it has been approved by the FDA under the brand name Olinvyk for intravenous use for treating severe acute pain when alternative therapeutic options are inadequate [166]. According to the prescribing information, the maximum cumulative daily dose should be 27 mg, and the period should be no longer than 48 h, because the sole approved use for oliceridine is to relieve postoperative pain through patient-controlled analgesia (PCA), i.v. infusion, or bolus. A 1.5 mg first dose given by a healthcare professional followed by 0.35 mg demand doses with a 6 min lockout could be used with supplemental doses of 0.75 mg starting 1 h after the original dose and continuing as needed after that. The patient is expected to gain pain relief in 2–5 min [167,174].

Preclinical studies have shown a higher potency of oliceridine than of morphine in achieving analgesia [132,175]. Notably, even at doses several times the maximum daily exposure of 40 mg/day, no oliceridine-induced toxicity other than the standard opioid side effects (such as decreased activity, lower blood pressure, and body temperature) and generally fewer adverse events were detected [132]. According to Liang and colleagues, the level of physical dependence produced by oliceridine is comparable to or no different from that produced by morphine, and compared to morphine, tolerance is less likely to develop during long-term oliceridine treatment. Additionally, the degree of sensitization or opioid-induced hyperalgesia produced by oliceridine is not as severe as that produced by morphine [175,176].

Ascending TRV130 doses were used in the first TRV130 in-human clinical trial to study the drug’s pharmacokinetics, pharmacodynamics, and tolerability in healthy volunteers. Nausea and vomiting were reported at the 7 mg dose, which prevented further dose escalation. Doses between 0.15 and 7 mg administered intravenously over 1 h were well tolerated [177].

Based on these findings, Soergel and colleagues conducted a randomized, double-blind, placebo-controlled, crossover study (NCT02083315) enrolling 30 healthy male volunteers. During the 11-day/10-night sequestration, the volunteers randomly received single doses of TRV130, a placebo, or morphine intravenously on days 1, 3, 5, 7, and 9. TRV130 plasma concentration measurements reached their peak within 10 min of infusion and then decreased biphasically, indicating a fast distribution followed by an elimination phase. When compared to placebo and morphine at 10 and 30 min, TRV130 at all dosages (1.5, 3, and 4.5 mg) caused a rapid and significant increase in CPT hand removal latency from baseline. This resulted in a temporary decrease in respiratory drive at all doses tested, which peaked at 30 min and was comparable in magnitude to that after morphine administration; however, unlike the transitory effect of TRV130, the effect of morphine on respiratory drive persisted through the final ventilatory response to the hypercapnia measurement at 4 h post dose. The drug TRV130 was generally well tolerated; however, more patients reported experiencing severe nausea after taking morphine than after TRV130. These adverse effects included nausea, dizziness, vomiting, headache, pruritus/flushing, and somnolence [131].

This evidence showed that oliceridine had a safer safety profile than morphine, which led to oliceridine having a clinical advantage over morphine within clinical concentration ranges; it is evident that at clinically relevant doses, oliceridine appears to separate analgesia from respiratory depression more effectively than morphine. However, the authors note that predicting respiratory depression occurrences based on the utility function should presently be regarded as exploratory, and requires additional research to support their findings [168].

Sixty healthy, non-dependent, recreational opioid users participated in a single-dose, randomized, double-blind crossover study to compare the abuse potential of intravenous oliceridine to that of intravenous morphine and placebo. The blinded treatments were given as 1 min intravenous infusions of oliceridine 1 mg, 2 mg, or 4 mg, morphine 10 mg or 20 mg, or placebo. Standard abuse liability endpoints were evaluated during each period. Based on the findings of a previous dose escalation trial (Phase A of trial 1011), it was anticipated that the effects of oliceridine 2 mg and 4 mg would be comparable to those of morphine 10 mg and 20 mg, respectively. The study found that oliceridine and morphine showed the same propensity for misuse at equianalgesic doses. For this reason, the Controlled Substances Act requires that oliceridine be classified as a Schedule II drug, which would provide the same regulations and safeguards as those now in place for traditional i.v. opioid drugs given in a hospital or controlled settings [173].

The pharmacokinetics of oliceridine was also investigated in a special population of patients. Simons and colleagues conducted a four-arm double-blind, randomized, crossover study that evaluated the respiratory effects of intravenous 0.5 or 2 mg oliceridine and 2 or 8 mg morphine in an elderly population (18 healthy male and female volunteers, aged 55 to 89), on four separate occasions. The results demonstrated that while at low doses of oliceridine, no significant respiratory effects were observed, the peak effects of respiratory depression from high doses of oliceridine and both doses of morphine occurred 0.5 to 1 h after opioid administration. However, oliceridine-induced respiratory depression recovered to baseline more quickly than morphine-induced respiratory depression. Moreover, the magnitude of the respiratory depression induced by oliceridine appeared smaller over time compared with that induced by morphine [178]. The assessment of oliceridine’s pharmacokinetics, tolerability, and safety in patients showing hepatic or renal impairment was performed in a phase I, open-label, single-dose clinical trial. The researchers compared the pharmacokinetics and safety of 0.5 mg of intravenous oliceridine to 1 mg in participants with end-stage renal disease (ESRD) or hepatic impairment. According to the results, neither oliceridine clearance nor AUC was impacted by hepatic impairment, nor was there a difference in clearance between persons with normal renal function and ESRD patients [179].

Phase II clinical trials have assessed the effectiveness and safety of several oliceridine doses and dosing regimens in comparison to placebo and a traditional intravenous opioid in managing moderate-to-severe pain after surgery.

A phase II, randomized, placebo- and active-controlled trial was conducted to examine the efficacy and tolerability of TRV130 for the treatment of acute pain after bunion surgery. This study was divided into two phases: stage A, or the pilot phase, in which 144 patients with moderate-to-severe acute pain following bunion surgery were randomized to receive double-blind TRV130 (1 mg, 2 mg, 3 mg, or 4 mg every 4 h), a placebo, or morphine (4 mg every 4 h); and stage B, in which 195 patients were randomized to receive double-dummy TRV130 (0.5 mg, 1 mg, 2 mg, or 3 mg every 3 h), placebo, or morphine 4 mg every 4 h intravenously. The oliceridine dosing interval was reduced and the doses were changed in accordance with the stage A results, since the dose-related decrease in pain intensity that was visible early in the oliceridine dosing interval was essentially absent by the conclusion of the dosing interval. The results showed that TRV130 at 2 and 3 mg produced significantly greater categorical pain relief than morphine after the first dose, with meaningful pain relief occurring in 5 min, and that morphine at 4 mg produced significant reductions in pain intensity than placebo over the course of 48 h. Additionally, it has been determined that after a single dose, oliceridine is almost five times more potent than morphine. No significant side effects were observed with TRV130, and it was tolerated similarly to morphine [180].

Another phase IIb, randomized, double-blind, patient-controlled analgesia trial (NCT02335294) compared oliceridine to morphine and placebo in terms of effectiveness, safety, and tolerability in patients experiencing moderate-to-severe pain after abdominal surgery (NCT02335294). A total of 200 patients were randomly assigned to one of three treatment regimens in the first stage of the study, which was divided into two phases. These treatment regimens were placebo, oliceridine (1.5 mg loading dose and 0.1 mg demand doses with a 6 min lockout interval), or morphine (4 mg loading dose and 1 mg demand doses). The interim analysis led to an increase in the oliceridine demand dose from 0.10 to 0.35 mg. According to efficacy assessments, oliceridine treatment resulted in a statistically significant analgesia when compared to placebo, and the analgesia it caused for the full 24 h of treatment was comparable to morphine. A notable difference was the speed at which the pain was relieved, which was statistically different from both the placebo and morphine. All treatments were generally well tolerated, showing a limited incidence of reported undesired effects, mainly nausea, vomiting, and headache, asserting a favorable safety and tolerability profile [181].

APOLLO-1 (NCT02815709) was a phase III, double-blind, randomized trial for individuals who had undergone bunion surgery and were experiencing moderate-to-severe pain. Similar to the prior study, PCA was used to administer the medication. The study enrolled 389 patients who were randomly assigned to one of five group medications: placebo; oliceridine 1.5 mg loading dose/0.1 mg on demand; oliceridine 1.5 mg loading dose/0.35 mg on demand; oliceridine 1.5 mg loading dose/0.5 mg on demand; and morphine 4 mg loading dose/1 mg on demand. All oliceridine dose regimens were shown to have considerably greater responder percentages than placebo as the primary endpoint, and the 0.35 and 0.5 mg dosing regimens were comparable to morphine. The prevalent side effects of oliceridine were similar to those of conventional opioids, and became stronger with increasing doses. While the highest demand dose regimen of oliceridine (0.5 mg) was not statistically distinct from morphine, patients in the 0.1 and 0.35 mg regimens had a considerably decreased risk of experiencing a respiratory safety event compared with morphine. The incidence of gastrointestinal side effects was also similar to morphine in the 0.5 mg dose schedule, but lower than morphine in the 0.1 and 0.35 mg treatment regimens [182].

The effectiveness and safety of oliceridine for acute pain after abdominoplasty were assessed in the APOLLO-2 (NCT02820324) trial. With respect to APOLLO-1, patients were given a loading dosage of either a placebo, oliceridine (1.5 mg), or morphine (4 mg), and then demanded doses through patient-controlled analgesia (0.1, 0.35, or 0.5 mg oliceridine, 1 mg morphine, or placebo) with a 6 min lockout interval. The investigated drug was administered to a total of 401 subjects. All oliceridine dosing regimens showed statistically superior analgesia to placebo, with a higher number of treatment responders in the 0.35 and 0.5 mg demand dose regimens across the 24 h treatment period, which is consistent with earlier findings in both phase II and phase III trials. The oliceridine 0.35 and 0.5 mg demand dose regimens were consistently superior to placebo in terms of the proportion of treatment responders over time, the amount and timing of self-reported pain relief, the proportion of patients receiving rescue pain medication over time, the timing of the first use of rescue pain medication, and clinician and patient-reported satisfaction. In comparison to morphine, oliceridine (0.35 mg regimen) had a better safety and acceptability profile in relation to respiratory and gastrointestinal side effects. Similar to earlier trials, the oliceridine 0.35 and 0.5 mg dose regimens seemed to reduce the respiratory safety burden (a measure to assess the risk of opioid-induced respiratory depression (OIRD)), albeit this effect was not statistically significant [183].

ATHENA was a phase III, multicenter, open-label clinical trial (NCT02656875) that aimed to represent intravenous opioid use in a large, “real world” setting by being less restrictive in terms of patient eligibility criteria, treatment protocol requirements, patient population, and mode of administration. Patients 18 years of age or older with moderate-to-severe acute pain after surgery or with a non-surgical painful medical condition were included. In total, 768 patients who had been enrolled received i.v. oliceridine treatment through PCA and/or bolus dosage provided by physicians. A loading dose of 1 to 2 mg for intravenous bolus dosing was given, and if necessary, a supplemental dose of 1 mg was given within 15 min. Every 1 to 3 h, as needed, additional doses of 1 to 3 mg were given. In the case of PCA, a 6 min lockout interval was used to provide a 1.5 mg loading dosage and a 0.5 mg demand dose. Supplementary dosages of 1 mg were permitted as soon as 15 min following the original dose, if clinically necessary. 76% of patients received local anesthetics, 69% received non-steroidal anti-inflammatory medicines (NSAIDs), and 48% received oral opioids prior to the first dosage of oliceridine. The findings of this study showed that oliceridine was generally safe and well tolerated in a large population whether delivered alone or as a component of multimodal analgesia to adult patients experiencing moderate-to-severe pain due to surgical procedures or medical disorders. In line with previous investigations, oliceridine was linked to a strong analgesic effect and quick onset of action. Although not free from side effects (64% of patients experienced adverse events while receiving oliceridine treatment), the side effects were generally mild and consistent with previous data. Notably, individuals receiving oliceridine had a much lower incidence of OIRD events compared with those receiving conventional opioids despite the inclusion of patients with risk factors for ORAEs, such as advanced age, obesity, and sleep apnea. The absence of a concurrent control group, however, is the study’s primary flaw [184].

Brzezinski and colleagues conducted an exploratory and retrospective examination of ATHENA data to learn more about the prevalence of OIRD in aged and/or obese surgical subjects. They discovered that using oliceridine for postsurgical analgesia in patients with advanced age and/or higher body mass index (BMI) who were experiencing moderate-to-severe pain was not linked to an increased risk of respiratory depression, and may even be clinically appropriate for this group of patients, who are at a high risk of OIRD [185].

A phase IV clinical trial (VOLITION, NCT04979247) which is presently enrolling 200 patients and is expected to be completed by July 2025 is investigating the effects of oliceridine. The trial’s design specifications state that patients will receive a bolus of 1.5 mg oliceridine near the conclusion of the operation. After surgery, PCA will be initiated with 0.35 mg demand doses, a 6 min lock-out, and no background infusion. Following the initial 1.5 mg loading dosage, additional boluses (1 mg) of oliceridine will be administered as soon as 15 min later, depending on the NRS score and clinical evaluation of the patient. A dose of 0.5 mg oliceridine can be added to the PCA demand dose in addition to additional 1 mg bolus doses. The primary goal of the clinical trial was to assess the percentage of participants who experienced respiratory compromise 24 h after receiving the first dose. Oliceridine currently has only one approved use, that is, relieving postoperative pain through PCA, IV infusion, or bolus. Evidence suggests that this drug has the potential to be used for additional therapeutic purposes in addition to serving as an important part of a multimodal analgesic regimen, particularly for people who have risk factors such as advanced age, obesity, and sleep apnea. Table 2 provides a thorough summary of the clinical trials involving oliceridine.

## 5. Conclusions and Future Perspectives

In summary, in this article, we have reviewed the current status of the use of opioids as therapeutic agents employed to treat several diseases, including cancer and non-cancer pain. We paid special attention to the analysis of the opportunity to target opioid receptors for treating opioid use disorder (OUD), drug withdrawal, and addiction. Although there is great pharmacological potential in targeting opioid receptors, the clinical use of opioids is severely limited by several adverse effects such as tolerance and physical and physiological dependence. Finally, we investigated some drugs targeting opioid receptors that seem to hold great promise for treating severe pain but could also be indicated for individuals suffering from OUD, such as mitragynine, MCAM, and oliceridine, analyzing experimental and clinical evidence. We also considered the significant improvements made in the last decade in biophysical techniques such as cryo-EM, which allowed us to obtain experimental three-dimensional structures of several members of the G-protein-coupled receptor family, including opioid receptors. In the future, it is expected that this structural information will be useful for designing a novel generation of opioid receptor ligands [186,187]. In fact, understanding the binding mode of agonists and antagonists related to reduced side effects could pave the way for the rational design of drugs able to target opioid receptors with improved efficacy and selectivity. Accordingly, in few years, it is expected that innovative opioid receptor-targeting drugs will be entered into clinical trials to provide clinically usable molecules for treating OUD, and that these drugs will later be employed to improve the quality of life of patients with chronic pain and related disorders.

## Figures and Tables

**Figure 1 ijms-24-10888-f001:**
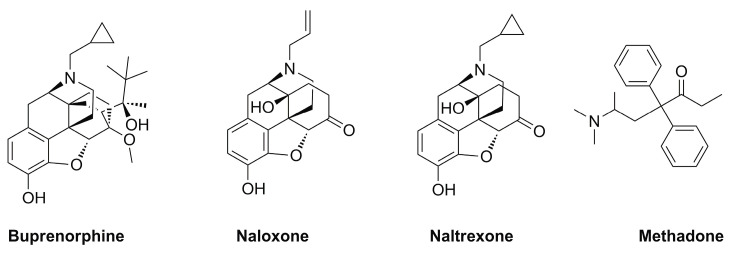
Chemical structures of the main drugs used in opioid replacement therapy (ORT).

**Table 1 ijms-24-10888-t001:** Opioid receptor types and their main roles in physiological and non-physiological functions.

Receptor	Main Functions	Tissue of Expression	Endogenous Ligand	Drugs ^a^
μ-opioid receptor μ_1;_ μ_2_; μ_3_	pain supraspinal analgesia reward euphoria respiratory depression gastrointestinal motility anorexia pruritus urinary retention physical dependence	brain sub-mucosal plexus mesenteric plexus spinal cord	endomorphins endorphins enkephalins	morphine (agonist) methadone (agonist) fentanyl (agonist) heroin (agonist) buprenorphine (partial agonist) nalmefene (inverse agonist) cebranopadol (agonist) naloxone (antagonist) naltrexone (antagonist)
κ-opioid receptor κ_1_; κ_2_; κ_3_	pain spinal analgesia sedation stress miosis dyspnea respiratory depression dysphoria anhedonia aversion dependence/addiction	brain mesenteric plexus peripheral sensory neurons	dynorphins	morphine (agonist) methadone (agonist) fentanyl (agonist) heroin (agonist) nalmefene (partial agonist) cebranopadol (partial agonist) naloxone (antagonist) buprenorphine (antagonist) naltrexone (antagonist)
δ-opioid receptor δ_1_; δ_2_	pain analgesia antidepressant/anxiolytic effects	brain mesenteric plexus spinal cord peripheral sensory neurons	met-enkephalin leu-enkephalin endorphins	methadone (agonist) fentanyl (agonist) heroin (agonist) cebranopadol (agonist) naloxone (antagonist) buprenorphine (antagonist) naltrexone (antagonist)
opioid receptor-like 1 (ORL1)	pain, inflammatory and neuropathic pain fibromyalgia	forebrain (cortical areas, olfactory regions, limbic structures, thalamus)	nociception/orphanin FQ (N/OFQ)	cebranopadol (partial agonist) buprenorphine (partial agonist)

^a^ the behavior of the ligand is reported according to the information featured on the DrugBank website (https://go.drugbank.com/, accessed on 21 January 2023).

**Table 2 ijms-24-10888-t002:** Clinical studies on oliceridine: an overview and findings.

	Study	Subjects	Dosing Regimen	Results	Adverse Events	Ref
**Phase I**	First-in-human study	74 healthy volunteers	Oliceridine i.v.: dose range 0.15 to 7 mg administered over 1 h Oliceridine i.v.: 1.5 mg administered as 30, 15, 5, and 1 min infusions	Dose and exposure-related pupil constriction, confirming central compartment µ-opioid receptor engagement Marked pupil constriction noted at 2.2, 4, and 7 mg doses	Nausea and vomiting observed at the 7 mg dose limited further dose escalation	[177]
	Single-center, randomized, double-blind, placebo-controlled, 5-period, crossover study	30 healthy men aged 18 to 50 years, with body mass indices of 19.0 to 32.0 kg/m^2^.	Oliceridine: single intravenous injections 1.5, 3, or 4.5 mg Morphine: single intravenous injection 10 mg Placebo	Oliceridine at all doses elicited a rapid and significant increase in cold pain test hand removal latency from baseline compared to placebo with peak efficacy at the first measurement 10 min post dose Oliceridine at 3 and 4.5 mg also significantly increased hand removal latency compared to morphine at 10 and 30 min, after which latency was similar to morphine Oliceridine produced a transient reduction in respiratory drive at all doses tested The reduction in respiratory drive after morphine was similar in magnitude to the peak effect of oliceridine at 30 min; the effect of morphine persisted	Generally well tolerated, with reported adverse events consistent with action at the μ-opioid receptor, including nausea, vomiting, dizziness, somnolence, pruritus/flushing, and headache. These effects appeared to be dose-related Oliceridine 4.5 mg adverse events similar to that of morphine but a greater incidence of nausea, dizziness, pruritus, and headache	[131]
	Abuse liability study Single-dose, randomized, double-blind crossover trial	60 healthy, non-dependent, recreational opioid users	Oliceridine i.v. 1 min infusion: 1 mg, 2 mg, or 4 mg Morphine i.v. 1 min infusion: 10 mg or 20 mg Placebo	Equianalgesic doses of oliceridine and morphine had similar abuse potential (oliceridine 2 mg would be similar to morphine 10 mg, and oliceridine 4 mg would be similar to morphine 20 mg) The more rapid reductions in mean drug liking scores with oliceridine compared with morphine are consistent with oliceridine’s shorter t½ and lack of active metabolites		
	Four-arm double-blind, randomized, crossover study	18 healthy male and female volunteers aged 55 to 89 years	Oliceridine i.v.: 0.5 or 2 mg 1 min infusion Morphine i.v.: 2 or 8 mg morphine 1 min infusion	High-dose oliceridine and high- and low-dose morphine showed a rapid drop in mean isohypercapnic ventilation, VE55, an indication of rapid onset of respiratory depression, within 30 min of administration High-dose oliceridine and low-dose morphine returned toward baseline within 3 h, and high-dose returned toward baseline in more than 6 h Low-dose oliceridine did not produce any significant respiratory depression	At every dose, the total number of events was similar between opioids The most frequently reported events were dizziness, lightheadedness, somnolence, and vertigo after oliceridine administration; and nausea, lightheadedness, dizziness, and somnolence following morphine	[178]
	Open-label, single-dose trial	51 subjects, males or females, aged 18–80 years, BMI 18.0–35.0 kg/m^2^, a minimum weight of 50 kg; controls, age- and sex-matched at a ratio of 1:1	Subjects with renal impairment received a single oliceridine 0.5-mg dose infused over 2 min Subjects with mild hepatic, moderate, or severe hepatic impairment received a dose of oliceridine 0.5 mg infused over 2 min Healthy subjects in both studies received a dose of oliceridine 1 mg infused over 2 min	No clinically meaningful difference in any PK parameter between ESRD and healthy subjects Clearance and dose-normalized AUC showed no change with the degree of hepatic impairment Dose-normalized Cmax was significantly lower in the severe hepatic- impairment group compared with the other groups. Half-life increased with the degree of hepatic impairment. The volume of distribution increased with the degree of hepatic impairment	6 of 17 subjects (35.3%) in the renal study experienced a total of 11 mild treatment emerged adverse events, mainly nausea, fatigue, and euphoria. They occurred more frequently in the 1-mg dose group compared with the 0.5-mg dose group In the hepatic impairment study only 2 adverse events of somnolence were reported in 2 subjects in the mild hepatic impairment group Oliceridine had no measurable impact on laboratory parameters, vital signs, ECG parameters, or oxygen saturation that could be attributed to hepatic impairment	[179]
**Phase II**	Randomized, placebo- and active-controlled trial	141 patients in the pilot phase, 192 patients in phase II, experiencing moderate-to-severe acute pain after bunionectomy	Oliceridine i.v. bolus: 1, 2, 3, or 4 mg i.v. q4h (Pilot Phase); 0.5, 1, 2, or 3 mg i.v. q3h (Phase 2) Morphine i.v. bolus: 4 mg q4h	Oliceridine 2 and 3 mg every 3 h, and morphine 4 mg every 4 h produced significant reductions in pain intensity than placebo over 48 h Oliceridine 2 and 3 mg produced significantly greater categorical pain relief than morphine after the first dose, with meaningful pain relief occurring in 5 min Oliceridine is approximately five times more potent than morphine after a single dose	Oliceridine produced no serious adverse events, with tolerability similar to morphine	[180]
	Randomized, double-blind, patient-controlled analgesia trial (NCT02335294)	200 patients, males or females (99%) aged 18–65 years who planned to undergo abdominoplasty without any additional collateral procedures were enrolled	Oliceridine PCA: two loading doses of 0.7 and 5 mg separated by 10 min that were followed by demand doses of 0.1 mg (A) or 0.35 mg (B) with a 6-minlockout interval Morphine PCA: two loading doses of 2 mg separated by 10 min that were followed by demand doses of 1 mg with a 6 min lockout interval	Oliceridine regimens A and B produced statistically significant reductions in pain relative to placebo, which was similar to that of morphine The proportion of patients using rescue analgesics was 31% with oliceridine regimen A, 21% with oliceridine regimen B, and 25% with morphine, compared with 64% with placebo	The most frequently reported events were nausea, vomiting, and headache Lower percentages of patients treated with oliceridine experienced nausea, vomiting or respiratory effects than patients receiving morphine No clinically significant changes from baseline were reported in vital signs	[181]
**Phase III**	APOLLO-1: multicenter, double-blind, randomized, placebo- and active-controlled trial (NCT02815709)	389 patients, males and females (84.8%), aged 18–75 years and scheduled to undergo primary, unilateral, first metatarsal bunionectomy	Oliceridine PCA: loading dose of 1.5 mg followed by demand doses of 0.1, 0.35, or 0.5 mg with a 6 min lockout interval Morphine PCA: loading dose of 4 mg followed by demand doses of 1 mg with a 6 min lockout interval	Percentage of treatment responders: 50, 62, and 65.8% (oliceridine) vs. 71.1% (morphine) Oliceridine 0.35 mg and 0.5 mg regimens provide pain relief comparable to the morphine regimen	The most common adverse events were nausea, vomiting, headache, dizziness, constipation, somnolence or sedation, pruritus, and dry mouth No patients experienced serious adverse events; few patients reported a severe adverse events The incidence of respiratory safety events and their duration increased in a dose-dependent manner across the oliceridine treatment groups	[182]
	APOLLO-2: double-blind, randomized, placebo- and active-controlled trial (NCT02820324)	401 patients, males and females (99.3%), aged 18–75 years, BMI < 35 kg/m^2^ or body weight > 40 kg, who followed abdominoplasty procedure with no additional collateral procedures	Oliceridine PCA: loading dose of 1.5 mg followed by demand doses of 0.1, 0.35, or 0.5 mg with a 6 min lockout interval Morphine PCA: loading dose of 4 mg followed by demand doses of 1 mg with a 6 min lockout interval Volume-matched placebo	Percentage of treatment responders: 61, 76.3, and 70% (oliceridine) vs. 78.3% (morphine) Higher proportion of responders in the 0.35- and 0.5-mg demand dose regimens over the 24-h In comparison with morphine, dose regimens of 0.35 and 0.5 mg oliceridine were considered equi-analgesic Rapid onset of effect with all oliceridine regimens compared to placebo at early time points, particularly at 10 and 15 min	The overall proportion of patients experiencing at least 1 adverse event was lowest with placebo (78.3%) and increased in a dose regimen–dependent manner across the oliceridine 0.1-, 0.35-, and 0.5-mg demand dose regimens (89.6%, 93.7%, and 95%, respectively). The proportion of patients experiencing at least 1 adverse event with morphine was 97.6%. These adverse events included mainly nausea, headache and hypoxia The proportion of patients experiencing gastrointestinal adverse events increased with higher oliceridine demand dose regimens, with the higher dose of 0.5 mg being similar to those observed with morphine Serious adverse events were reported in 5 patients (4 patients in the oliceridine treatment regimens and 1 patient in the morphine treatment regimen). Such as post-procedural hemorrhage, syncope, and lethargy that were reported with the 0.5-mg regimen, and abdominal wall hematoma with the 0.35-mg regimen	[183]
	ATHENA: multicenter, open-label trial (NCT02656875)	768 patients, males and females (65%), >18 years, Caucasian (78%), surgical and non-surgical patients with painful medical conditions	Oliceridine i.v. bolus: loading dose of 1 to 2 mg and a supplemental dose of 1 mg within 15 min if needed, followed by doses of 1 to 3 mg every 1 to 3 h as needed Oliceridine PCA: loading dose of 1.5 mg and demand doses of 0.5 mg with a 6 min lockout interval, and supplemental doses of 1 mg could be given as needed	Rapid reduction in pain intensity (2.2-point reduction of pain score within 30 min) which were comparable across cumulative oliceridine dose groups In patients with follow-up assessment available at their end of treatment (n = 225), data indicate maintenance of pain reduction Lack of efficacy leading to discontinuation was reported in less than 5% of patients	Most adverse events were of mild (37%) or moderate (25%) intensity, mainly nausea (31%), constipation (11%), and vomiting (10%) Only 3 patients experienced serious adverse events SAEs possibly related to oliceridine: post-operative ileus (1 patient), respiratory depression with respiratory rate <8 breaths/min within 5 h of receiving oliceridine (1 patient), hepatic and renal failure confounded by surgical-related complications (1 patient) The incidence of adverse events in obese patients (defined as BMI ≥ 30 kg/m^2^) was 61% (vs. 64% in the overall population); the incidence of adverse events in the elderly patients was similar in the elderly overall population (around 30%)	[184]
**Phase IV**	VOLITION: A multicenter pilot cohort study	200 patients, males and females ≥18 years	Oliceridine i.v. bolus: loading dose of ≤1.5 mg near the end of surgery supplemental dose of ≤1 mg, 15 min after the initial dose Oliveridine PCA: demand doses of 0.35 mg (6 min lockout) increased to 0.5 mg if necessary	In progress Primary outcome: number of patients who have respiratory compromise Secondary outcome: cumulative duration of oxygen saturation < 90% and cumulative duration of respiratory rate < 8 breaths/min	In progress	NCT04979247

## Data Availability

Not applicable.

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
