# Peer review of "Targeting Opioid Receptors in Addiction and Drug Withdrawal: Where Are We Going?"

_ijms, 2023, doi:10.3390/ijms241310888_

Round 1
Reviewer 1 Report
Overall, the text provides a comprehensive overview of opioid addiction and opioid use disorder (OUD), discussing the prevalence, diagnosis, treatment applications, side effects, and the need for the development of novel opioid agents. Here are some suggestions for improving the text:
Although the text briefly mentions the involvement of the locus ceruleus in opioid withdrawal and the role of adenylate cyclase, no other neurotransmitter systems involved in the withdrawal process are discussed. It would be useful to discuss how dysregulation of other neurotransmitters, such as glutamate and serotonin, can contribute to withdrawal symptoms.
The text focuses predominantly on the pharmacological aspects of the functioning of opioid receptors and treatments for OUD. By discussing the role of genetic, environmental, and psychosocial factors in the etiology of OUD, a more comprehensive understanding of the disorder could be attained.
The text emphasizes the need for novel opioid receptor ligands to minimize adverse effects while treating OUD. However, it does not clearly explain why these novel ligands are anticipated to have reduced side effects. In addition, the review would benefit from discussing alternative, non-opioid-based treatments for OUD and pain management, such as the development of medications targeting other neurotransmitter systems or non-pharmacological approaches such as cognitive-behavioral therapy.
The discussion of buprenorphine, methadone, and naltrexone as OUD treatments could be expanded to include a comparison of their relative efficacy and safety profiles, as well as any contraindications or patient populations for whom one treatment may be more suitable than the others.
Given that many individuals with OUD do not receive adequate care and relapse rates are high even among those who do receive treatment, it would be beneficial to discuss the challenges and potential strategies for improving access to and adherence with OUD treatments.
The text briefly mentions psychosocial counseling as a treatment adjunct to pharmacological therapy for OUD, but does not elucidate on the various types of counseling or behavioral interventions that may be advantageous. Including a more in-depth discussion of the function of interventions such as Cognitive Behavioral Therapy (CBT), Contingency Management, and 12-Step Facilitation Therapy in the management of OUD would provide a more complete picture of treatment options.
The significance of harm reduction strategies in the management of OUDs could also be discussed, as these methods are crucial for minimizing the negative effects of drug use. Harm reduction strategies include syringe exchange programs, supervised injection facilities, and education on overdose prevention.
The potential function of drug-drug interactions in the management of OUD is not discussed in the text. Understanding and addressing the potential for interactions between OUD treatments and other medications is essential for the safe and effective management of patients with OUD who may be taking multiple medications for various conditions.
The review would benefit from a more in-depth discussion of the difficulties associated with identifying and diagnosing OUD, as well as the obstacles to obtaining appropriate treatment. Some of these obstacles include stigma, a dearth of healthcare professionals trained in addiction medicine, and the limited availability of evidence-based remedies in certain regions.
The review could conclude with a discussion of the significance of a comprehensive and integrated approach to OUD management, including medical, psychological, and social support services to address the complex requirements of individuals with OUD. This strategy may aid in enhancing treatment outcomes and promoting long-term recovery.
Minor editing of English language required
Author Response
Overall, the text provides a comprehensive overview of opioid addiction and opioid use disorder (OUD), discussing the prevalence, diagnosis, treatment applications, side effects, and the need for the development of novel opioid agents. Here are some suggestions for improving the text:
Authors: we thank the referee for the comments for improving the manuscript. We addressed the points raised by the reviewers.
Although the text briefly mentions the involvement of the locus ceruleus in opioid withdrawal and the role of adenylate cyclase, no other neurotransmitter systems involved in the withdrawal process are discussed. It would be useful to discuss how dysregulation of other neurotransmitters, such as glutamate and serotonin, can contribute to withdrawal symptoms.
Authors: according to the reviewer suggestion we briefly reported the dysregulation of other neurotransmitters (dopamine, noradrenaline, serotonin, and glutamate) in withdrawal process from line 127.
The text focuses predominantly on the pharmacological aspects of the functioning of opioid receptors and treatments for OUD. By discussing the role of genetic, environmental, and psychosocial factors in the etiology of OUD, a more comprehensive understanding of the disorder could be attained.
The text emphasizes the need for novel opioid receptor ligands to minimize adverse effects while treating OUD. However, it does not clearly explain why these novel ligands are anticipated to have reduced side effects. In addition, the review would benefit from discussing alternative, non-opioid-based treatments for OUD and pain management, such as the development of medications targeting other neurotransmitter systems or non-pharmacological approaches such as cognitive-behavioral therapy.
Authors: please check the last response in which we highlighted that for addressing the mentioned points we introduced a new chapter (3. Psychosocial interventions for substance abuse and dependence). We also reported a brief discussion on the role and possible targeting of other neurotransmitter systems.
The discussion of buprenorphine, methadone, and naltrexone as OUD treatments could be expanded to include a comparison of their relative efficacy and safety profiles, as well as any contraindications or patient populations for whom one treatment may be more suitable than the others.
Authors: according to the suggestion we have expanded the discussion on buprenorphine, methadone, and naltrexone adding details about efficacy and comparing the results found for the mentioned drugs in the introduction.
Given that many individuals with OUD do not receive adequate care and relapse rates are high even among those who do receive treatment, it would be beneficial to discuss the challenges and potential strategies for improving access to and adherence with OUD treatments.
The text briefly mentions psychosocial counseling as a treatment adjunct to pharmacological therapy for OUD, but does not elucidate on the various types of counseling or behavioral interventions that may be advantageous. Including a more in-depth discussion of the function of interventions such as Cognitive Behavioral Therapy (CBT), Contingency Management, and 12-Step Facilitation Therapy in the management of OUD would provide a more complete picture of treatment options.
The significance of harm reduction strategies in the management of OUDs could also be discussed, as these methods are crucial for minimizing the negative effects of drug use. Harm reduction strategies include syringe exchange programs, supervised injection facilities, and education on overdose prevention.
Authors: please check the last response in which we highlight that for addressing the mentioned point we introduced a new chapter (3. Psychosocial interventions for substance abuse and dependence).
The potential function of drug-drug interactions in the management of OUD is not discussed in the text. Understanding and addressing the potential for interactions between OUD treatments and other medications is essential for the safe and effective management of patients with OUD who may be taking multiple medications for various conditions.
Authors: we discussed this issue on drug-drug interactions related to OUD from line 247
The review would benefit from a more in-depth discussion of the difficulties associated with identifying and diagnosing OUD, as well as the obstacles to obtaining appropriate treatment. Some of these obstacles include stigma, a dearth of healthcare professionals trained in addiction medicine, and the limited availability of evidence-based remedies in certain regions.
The review could conclude with a discussion of the significance of a comprehensive and integrated approach to OUD management, including medical, psychological, and social support services to address the complex requirements of individuals with OUD. This strategy may aid in enhancing treatment outcomes and promoting long-term recovery.
Authors: we thank the referee for the valuable suggestions. According to the request about the comprehensive approach to treat OUD, we inserted a novel chapter (3. Psychosocial interventions for substance abuse and dependence), that enclosed all the points that are requested to treat by the reviewer. In particular, we tried to address all the points raised by the referee, including a discussion of the function of interventions such as Cognitive Behavioral Therapy (CBT), Contingency Management, and 12-Step Facilitation Therapy in the management of OUD as well as the significance of harm reduction strategies. If the corrections done are not sufficient to address the issues, we will be ready to provide further corrections.
Reviewer 2 Report
The manuscript as submitted has significant errors in content (e.g. claiming that opioid antagonists are not opioid), has complicated and confusing sentence structure, and is not at all focused or concise enough to be a worthwhile addition to the field.
The authors seem to lack basic understanding of pharmacology (eg agonist vs antagonist, how affinity might effect the ability of a compound to block the effects of an abused opioid, duration of action and how that might effect dosing, efficacy vs potency, receptor reserve, etc.) which makes many of their statements incorrect or misleading.
Many of the word choices were odd/confusing and the grammar made many sentences and paragraphs difficult to follow. It is unclear if this is a language barrier problem or just poor editing.
Author Response
The manuscript as submitted has significant errors in content (e.g. claiming that opioid antagonists are not opioid), has complicated and confusing sentence structure, and is not at all focused or concise enough to be a worthwhile addition to the field.
The authors seem to lack basic understanding of pharmacology (eg agonist vs antagonist, how affinity might effect the ability of a compound to block the effects of an abused opioid, duration of action and how that might effect dosing, efficacy vs potency, receptor reserve, etc.) which makes many of their statements incorrect or misleading.
Authors: we thank the referee for the comments. In order to avoid possible mistakes about the mentioned issues we profoundly revised the paper. If the corrections done are not sufficient to address the issues, please provide more detailed information about the errors you found. We will be ready to provide further corrections.
Reviewer 3 Report
This review highlights a series of novel compounds including the opioid oliceridine recently approved by FDA, as drugs for the prevention and therapy of addiction and drug withdrawal.
This is a timely and important review, especially for researchers of opioid receptor studies in IJMS readers.
1. There are some typos and others that should be corrected.
1) L. 389
3.1.2 Mitragynine (Kratom) . It would be cited both.
2) P. 7-8, L.266-324
Nomenclatures are m-, d- and k-receptors among opioid receptors in the ms.
Change mu, delta, and kappa to the above abbreviation (L. 285, 289, … 306, 324)
Also, change m opioid receptors to m-opioid receptors. (L. 324)
3) L.301, What is DOA?
L.309, Change opioid use disorder to OUD.
4) P.11, L.480
In conclusion, mitragynine (kratom), one of the …
2. References
There is a literature that appeared in 2020 regarding structures, its site of action, and clinical studies of TRV130 (oliceridine). It should be quoted in the Introduction or Section 3.2
none
Author Response
This review highlights a series of novel compounds including the opioid oliceridine recently approved by FDA, as drugs for the prevention and therapy of addiction and drug withdrawal. This is a timely and important review, especially for researchers of opioid receptor studies in IJMS readers.
- There are some typos and others that should be corrected.
Authors: we thank the referee for the positive evaluation of the paper, we carefully checked the paper for avoiding typo errors.
1) L. 389 3.1.2 Mitragynine (Kratom) . It would be cited both.
Authors: we thank the referee for the suggestions, we revised the paper accordingly.
2) P. 7-8, L.266-324 Nomenclatures are m-, d- and k-receptors among opioid receptors in the ms.
Change mu, delta, and kappa to the above abbreviation (L. 285, 289, ... 306, 324) Also, change m opioid receptors to m-opioid receptors. (L. 324)
Authors: we thank the referee for the careful reading, we revised the nomenclature to avoid possible discrepancies.
3) L.301, What is DOA?
L.309, Change opioid use disorder to OUD.
4) P.11, L.480 In conclusion, mitragynine (kratom), one of the ...
Authors: we thank the referee for the suggestions, we revised the mentioned terms accordingly. DOA stands for duration of action, while we added mitragynine as indicated.
- References
There is a literature that appeared in 2020 regarding structures, its site of action, and clinical studies of TRV130 (oliceridine). It should be quoted in the Introduction or Section 3.2
Authors: as suggested we inserted the reference indicated by the reviewer.
Reviewer 4 Report
This manuscript is poorly written and therefore hard to follow. In my opinion it is also too long, especialy the part about oliceridine.
The authors sometimes use the word opiate , sometimes opioid. I think they should explain the difference or use only opioids. Opiats are natural compounds like morphine and derivatives. Opioids are used to describe endogenous peptides but they can also be used to describe morphine, so opioids is a wider concept.
Introduction; I do not understand what is opium poppy resin?
line 26 - 16 million are interested from this disorder - rather suffer.
line 73 When mentioning endogenous opioids the Authors omitt endomorphins. Endomorphin-1 and endomorphin-2 discovered by Zadina in 1997 are considered endogenous ligands of the mu receptor which is the one mostly engaged in pain relief.
line 70 to 90 - very unclearly described mode of action of GPCRs. Which units separate?
Table 1 As mentioned above, prefered ligands of the mu receptor are EM-1 and EM-2, not endorphins. Endorphins have equal affinity to mu and delta and are mainly responsible for mood. What means "data" in the last lines? The last column- I do not like the term exogenous ligand. There are a lot of exogenous ligands selective for each opioid receptor but they are not mentioned in this column. What we have there are drugs and the same are mentioned for all three receptors, though they mainly activate mu receptor.
lines 118-129 very unclear, hard to understand.
line 120 -what does it mean buprenorphine followed by naloxone in brackets? Buprenorphine is an agonist, naloxone an antagonist of opioid receptors. Why is it in brackets?
line 121 canceled not cancealed
line 159 It says that naltrexon is not a classical opioid. What do the authors understand by "classical opioids"? Usually, by classical opioids we understand endogenous peptides with Tyr-Gly-Phe-Phe- sequence at the N-terminus. Here the authors obviously have something else in mind, it would be better not to use the word classical.
line 278 HEK cells expressing opioid receptors - which?
Line 765 "biophysical techinques allowed the experimental solution of several members of the GPCR family.." - this sentence does not make sense.
Concluding, this manuscript may have some value as a review but not in its present form. English is unacceptable and there are several serious mistakes and discrepancies.
Poor quality of English.
Author Response
This manuscript is poorly written and therefore hard to follow. In my opinion it is also too long, especialy the part about oliceridine.
Authors: we thank the referee for the suggestion, we significantly shortened the section.
The authors sometimes use the word opiate , sometimes opioid. I think they should explain the difference or use only opioids. Opiats are natural compounds like morphine and derivatives. Opioids are used to describe endogenous peptides but they can also be used to describe morphine, so opioids is a wider concept.
Authors: we thank the referee for the suggestion. Accordingly, we replaced opiate(s) with opioid(s).
Introduction; I do not understand what is opium poppy resin?
Authors: commonly for opium poppy resin is intended the resin obtained from the Papaver somniferum. According to the comment, the text was revised in order to better understand the sentence removing the term.
line 26 - 16 million are interested from this disorder - rather suffer.
Authors: we revised the sentence accordingly
line 73 When mentioning endogenous opioids the Authors omitt endomorphins. Endomorphin-1 and endomorphin-2 discovered by Zadina in 1997 are considered endogenous ligands of the mu receptor which is the one mostly engaged in pain relief.
Authors: we apologize for the mistake, we added endomorphins as requested.
line 70 to 90 - very unclearly described mode of action of GPCRs. Which units separate?
Authors: we provided additional details on the mode of action of GPCRs in order to better understand the paragraph
Table 1 As mentioned above, prefered ligands of the mu receptor are EM-1 and EM-2, not endorphins. Endorphins have equal affinity to mu and delta and are mainly responsible for mood. What means "data" in the last lines? The last column- I do not like the term exogenous ligand. There are a lot of exogenous ligands selective for each opioid receptor but they are not mentioned in this column. What we have there are drugs and the same are mentioned for all three receptors, though they mainly activate mu receptor.
Authors: we thank the referee for the observation. In order to address the point, we revised the Table accordingly. We remove data, inserting the tissue of expression for the opioid receptor-like 1 (ORL1).
lines 118-129 very unclear, hard to understand.
Authors: we rewrote the sentence for a better reading.
line 120 -what does it mean buprenorphine followed by naloxone in brackets? Buprenorphine is an agonist, naloxone an antagonist of opioid receptors. Why is it in brackets?
Authors: we revised the sentence keeping only naloxone as antagonist.
line 121 canceled not cancealed
Authors: we revised the term.
line 159 It says that naltrexon is not a classical opioid. What do the authors understand by "classical opioids"? Usually, by classical opioids we understand endogenous peptides with Tyr-Gly-Phe-Phe- sequence at the N-terminus. Here the authors obviously have something else in mind, it would be better not to use the word classical.
Authors: we agree with the reference comment. For avoiding possible mistake, we removed the term classical as suggested.
line 278 HEK cells expressing opioid receptors - which?
Authors: we revised the sentence inserting the term mu-opioid receptor.
Line 765 "biophysical techniques allowed the experimental solution of several members of the GPCR family.." - this sentence does not make sense.
Authors: we revised the sentence accordingly.
Concluding, this manuscript may have some value as a review but not in its present form. English is unacceptable and there are several serious mistakes and discrepancies.
Authors: we thank the reviewer for her/his valuable comments. We revised the paper taking into account the suggestions, that allowed us to improve the paper. Furthermore, the English was carefully revised. If the corrections done are not sufficient to address the issues, we will be ready to provide further corrections.
Round 2
Reviewer 2 Report
Still not a significant contribution and poorly organized.
Odd word choices make it difficult to understand the meaning. Sentences are convoluted.
Reviewer 4 Report
The manuscript has been corrected according with my suggestions. English has been improved. In its present form the manuscript is suitable for publication.